# Evaluating the High-Temperature Properties and Reaction Mechanism of Terminal Blend Rubber/Nano Silica Composite Modified Asphalt Using Activated Rubber

**DOI:** 10.3390/nano12244388

**Published:** 2022-12-09

**Authors:** Nonde Lushinga, Zejiao Dong, Liping Cao

**Affiliations:** 1Department of Road and Railway Engineering, School of Transportation Science and Engineering, Harbin Institute of Technology, Harbin 150090, China; 2Department of Construction Economics and Management, School of Built Environment, The Copperbelt University, Kitwe P.O. Box 21692, Zambia

**Keywords:** terminal blend (TB) rubberized asphalt, nano silica, microwave-activated crumb rubber asphalt, storage stability, rheology

## Abstract

Terminal blend (TB) rubberized asphalt is a popular technology in the production of rubberized asphalt. However, it always presents challenges regarding the inadequate high-temperature rutting performance of the binders. Additionally, crumb rubber (CR), a modifier of asphalt is a cross-linked material which presents poor compatibility between CR particles and bitumen. Incorporating nanomaterials and pretreating CR particles are two possible solutions to address this drawback. But the performance improvement and modification mechanism of the composite TB binders is not clearly understood. Therefore, the purpose of this research was to evaluate the high-temperature properties and reaction mechanism of the TB rubber/nano silica composite modified asphalt using microwave activated rubber. To achieve the research purpose, bitumen penetration grade 80–100 was first modified with 8% CR particles at elevated temperature to produce TB rubberized asphalt followed by the addition of 0.5, 1.5 and 3.0% weight percentage of nano silica to produce TB rubber/nano silica composite modified asphalt. Short and long-term aging tests were performed on samples by thin film oven test (TFOT) and pressure aging vessel (PAV) prior to chemical and rheological tests. The results of the study shows that nano silica has a great influence on the high temperature rutting resistance, storage stability and anti-aging properties of TB rubberized asphalt. Nano silica promoted good interaction and compatibility between CR particles and bitumen and improved the overall rheological properties of the binders. XRD test results revealed that the TB rubberized/nano silica composite modified asphalt samples were amorphous materials and did not have a crystalline structure. The reaction mechanism between rubber and asphalt was found to be physical, whereas nano silica interacted chemically with TB rubberized asphalt. In light of these findings, this research concluded that nano silica evidently improves the high-temperature rutting properties of TB rubberized asphalt, which deserves further exploration and application.

## 1. Introduction

Waste tire accumulation has become a global environmental issue [1,2,3]. According to estimates, approximately one billion tires are made worldwide each year [4]. Crumb rubber (CR) made from waste tires is commonly used in asphalt modification. Studies have shown that the use of CR particles in the asphalt industry has a number of advantages, including increased service life, lower noise levels by up to 70%, improved thermal characteristics and skid resistance, and a safe technique for scrap tire recycling [5,6]. The production of crumb rubber modified asphalt (CRMA) has two technical routes. The first route is the “dry process”, in which crumb rubber modifier (CRM) is used to replace 1% to 3% of the aggregate weight in the asphalt mixture. In the dry process, CRM and asphalt have little interaction during mixing in the mix plant hence is not the most preferred. The second route is the “wet technique”, which was devised by McDonald in the late 1960s. In this route, CRM is added to the bitumen or base asphalt as a modifier at 160 °C and eventually heated at 180~190 °C [7]. The absorption of aromatic oils from asphalt cement into CRM is the predominant interaction between CRM and asphalt in the wet technique. According to Heitzman [8], interaction between CRM and asphalt is a physical reaction, not a chemical reaction. The “wet process” has a long record of use and has the potential to significantly improve results. However, due to a lack of storage stability, thorough quality control of CRMA is difficult [9].

According to the existing literature, there are several approaches for enhancing the storage stability of CRMA produced using the “wet process”. One of the most effective methods is to use terminal blend (TB) asphalt [10,11]. In this approach, CRMA is made at a high temperature and cured for a long time (more than four hours) to ensure that the CRM degradation is significant. TB asphalt, as with all other polymer modified asphalts, is made at the refinery (or terminal). The reaction conditions of TB binder are considerably different from those of CRM binder, making it more suitable for factory production. In TB asphalt, crumb rubber, loses its high-temperature anti-rutting properties when it is completely degraded [12]. Rutting is recognized and employed as the initial failure mechanism and one of the most critical factors in the design of flexible pavements. Permanent deformation of the wheel path in the horizontal direction emerges on the longitudinal surfaces, reducing the efficiency of the pavement and making vehicles rough and unsafe. Bitumen properties are important, to improve the rutting resistance of asphalt pavements. As a result of this, several researchers are now exploring the use of nanomaterials to improve the high-temperature properties and enhance the performance of TB binders [13], but the performance improvement and modification mechanism is not clearly understood. Additionally, it is worth noting that composite modified asphalt is an important solution for addressing the performance imbalance of single polymer modified asphalt [4]. Generally, TB binder has a higher storage stability than asphalt rubber (AR) since the asphalt completely digests the crumb rubber particles [12,14,15,16]. Recently, the Federal Highway Administration in collaboration with the University of California, Berkeley conducted accelerated pavement tests and found that TB asphalt outperformed AR in terms of fatigue resistance and is suitable for preparation of densely graded mixes [17,18]; however, the inferior rutting of TB binders is still a source of concern [19]. Lin et al. observed that light components such as aliphatic are generated during the manufacturing process of TB asphalt due to the desulfurization and degradation of rubber powder, which boosts TB asphalt’s low-temperature characteristics [20]. 

Besides the inferior high-temperature rutting properties of TB binders, it is a well-known fact that crumb rubber (CR) has the stereo-network structures established by cross-linking. Therefore, it is difficult to swell and distribute in asphalt matrix [6]. In this light, the surface activation of CR, which attempts to change the chemical and/or physical properties of the CR surface, is one possible solution to this challenge. Surface activation boosts the surface activity of CR particles, resulting in a strong interfacial adhesive ability between crumb rubber and asphalt matrix, and hence improves the properties of CRMA binders. In this context, microwave irradiation has emerged to be one of the economical methods of surface activation of CR particles. Surface activation treatment promotes the interaction between bitumen and crumb rubber particles [6]. In this research, CR particles were microwave irradiated to desulfurize and depolymerize the rubber before being added to base asphalt. The surface vulcanization network of CR particles broken by microwave irradiation enhances surface activity and, as a result improves bitumen compatibility [19]. According to Aoudia et al., the microwave irradiation of CR particles can disrupt the C-S and S-S bonds while keeping the C-C bond intact [21].

The purpose of this paper was to evaluate the high-temperature properties of terminal blend rubber/nano silica composite modified asphalt using microwave-activated rubber particles as well as elucidate its reaction mechanism. The effect of nano silica modifier on reducing high temperature rutting failure and its reaction mechanism of TB rubberized asphalt was evaluated and compared to the control sample using laboratory experiments. The research was beneficial from the standpoint of improving the inferior rutting performance of the TB rubberized asphalt as well as addressing the compatibility issues of CR particles and bitumen for sustainable paving solutions.

## 2. Materials and Experimental Procedures

### 2.1. Materials

In this study, the bitumen penetration grade 80–100 provided by Beijing Lusheng Asphalt Co., Ltd. was used as the base binder. The fundamental properties of bitumen are presented in Table 1. The 40-mesh CR produced from waste tires in which metals and fibers were removed was supplied by ZhongNeng Rubber Co. Ltd. Chengdu, Sichuan, China. The fundamental properties of CR are provided in Table 2. The fumed silica particles (Aerosil R202), herein referred to as nano silica, was obtained from Evonik Industries, China with the fundamental properties provided in Table 3.

### 2.2. Microwave Treatment of Crumb Rubber Particles

Microwave irradiation has emerged to be one of the economical methods of surface activation of CR particles. In this study, a domestic microwave with a frequency of 2450 MHz was used. Exactly 60 g of CR sample was put in a microwave beaker and microwave irradiated for 4 min. Prior to microwave irradiation, CR was dried at 60 °C to reduce its moisture content. The resulting sample was known as the Microwave-activated crumb rubber (MCR).

After the activation of crumb rubber particles, the surface topography of MCR was examined in scanning electron microscope (SEM). The results revealed that MCR sample was porous and loose, whereas those of unactivated crumb rubber (CR) was smooth and dense, as shown in Figure 1. The loose and porous surface increases the reaction area of CR with asphalt binders because it decreases the density of CR particles; hence, the reaction area is extensive. On the other hand, the unactivated crumb rubber is flat and certainly not good enough for dispersion [22]. Therefore, MCR modifier which would give better stability when blended with asphalt was employed in this research.

### 2.3. Asphalt Binder Modification

The terminal blend (TB) rubber–nano silica modified composite asphalt was prepared using a technique suggested by Han et al. [19]. Herein, base asphalt was heated to 160 °C and blended with 8% MCR (by weight of asphalt). After the addition of MCR into base asphalt, the temperature was increased to 180~190 °C to promote the mixing and swelling of the blend while stirring manually for 20~30 min. The MCR/asphalt blend was then sheared at 3000~5000 rpm for 40~50 min at 180~190 °C. This was followed by the addition of 0.5, 1.5 and 3.0% of nano silica by weight of asphalt to produce TB rubber/nano silica composite modified asphalt. The composite mix was sheared at 3000~5000 rpm for another 40~50 min at 180~190 °C. The modification procedure is schematically presented in Figure 2, and the labels used in this research are presented in Table 4.

Following the preparation of terminal blend (TB) rubber/nano silica composite asphalt binders, various laboratory experiments were conducted to study the influence of nano silica on properties of TB binders. The experimental flow chart employed herein is shown Figure 3.

### 2.4. Aging Procedure

One of the most important elements affecting the lifespan of an asphalt pavement is the aging of bituminous binder. Aging causes chemical and/or physical property changes in bituminous materials, making them harder and more brittle, increasing the likelihood of pavement failure. Bitumen aging is often divided into two stages: short-term aging at high temperatures during asphalt mixing, storage, and laying, and long-term aging at ambient temperatures while in service.

To simulate the short-term aging of asphalt binders in the laboratory, thin-film oven test (TFOT) was performed following ASTM D1754 standard [23]. According to this test, the samples were held at 163 °C in the Thin-Film Oven for the TFOT test. The samples were aged for 20 h using the PAV standard technique (300 psi, 100 °C). After the TFOT test, the asphalt samples were placed in a pressure aging vessel (PAV) at 100 °C and 2.1 MPa pressure for 20 h during the long-term aging tests.

### 2.5. Segregation Test

A laboratory approach for testing the likelihood of polymer to separate from polymer modified asphalt under static heated storage conditions is known as a segregation test. Testing on material prepared according to this approach can be used as a guideline for creating goods or establishing field handling guidelines. Large discrepancies in test findings between top and bottom specimens suggest that the polymer and the base asphalt are incompatible.

In this research, exactly 50 g of each TB asphalt binder was put into standard aluminum tubes 25 mm diameter by 125 mm to 140 mm length, as per the Standard Practice for Determining the Separation Tendency of Polymer from Polymer Modified Asphalt, ASTM D 7173 [24]. Then, the samples were maintained in the vertical vessel of an oven for 48 h at 163 °C. Thereafter, the samples were cooled in the refrigerator for 4 h after which three equal portions or parts were cut from the tube. The bottom and top parts of the modified bitumen samples were then separated and subjected to MSCR testing to determine the degree of separation or segregation index (SI).

### 2.6. Fourier Transport Infrared (FTIR) Test

FTIR spectroscopy is a non-dispersive and non-destructive technique of infrared (IR) spectroscopy that is widely used to investigate chemical functional groups in materials. The FTIR approach was created to address the limitations of dispersive infrared spectrometers, which only measure the intensity of a spectrum across a restricted range of wavelengths at a time. The dispersive infrared spectrometer uses a prism or grating to isolate distinct frequencies of energy produced from an IR source.

The FTIR test was undertaken in order to identify additives in modified binders and determine the modification mechanism. The modified bitumen was heated to flow at 160 °C before being placed on glass slides for the sample processing. The FTIR spectra of control and nano silica modified bitumen samples were obtained using a Thermo Fisher Scientific NICOLET-iZ10 FTIR spectrometer (Thermo Fisher Scientific, Waltham, Massachusetts, USA). The FTIR test wavenumber ranged between 4000 and 400 cm^−1^.

### 2.7. Atomic Force Microscopy (AFM) Test

AFM is a popular surface investigation tool for micro/nanostructured coatings. It provides both qualitative and quantitative data on a variety of physical attributes, such as size, morphology, surface texture, and roughness, among others. In this research, the homogeneity of asphalt samples modified with or without nano silica was obtained using a Bruker Dimension ICON2-SYS AFM instrument (Bruker Corporation, Billerica, MA, USA). The asphalt samples were poured on glass slides and allowed to flow. Further details about the preparation of AFM samples are described elsewhere [25].

### 2.8. X-ray Diffraction (XRD) Test

The materials science technique of X-ray diffraction analysis (XRD) is used to determine the crystallographic structure of a material. XRD is a method of irradiating a substance with incoming X-rays and then measuring the intensities and scattering angles of the X-rays that depart the substance. In this study, The XRD test was conducted using X’Pert PRO MPD (*λ* = 1.54 A, K*α* Ratio = 0.5, voltage = 40 kV) from PANalytical Co. (Armsterdam, The Netherlands) with CuK*α* a radiation. The diffraction pattern was collected in 2 h with a step of 0.025 in the range of 10 to 80. Each sample was flattened into a tiny tablet with a thickness of about 2 mm, weighing exactly 3 g. The tests were conducted out at a room temperature of 25 degrees Celsius. The patterns were used to investigate the presence of nano silica in the asphalt matrix and its dispersion.

### 2.9. Frequency Sweep (FS) Test

A frequency sweep test is conducted to measure the dynamic modulus and phase angle of the binder by subjecting the sample to oscillation shear loading and varying the loading frequency. In this research work, the FS image test was performed, utilizing the state-of-the-art DHR-2 manufactured by T.A. Instruments (New Castle, DE, USA). Therein, a 25 mm diameter by 1 mm thick and 8 mm diameter by 2 mm thick formed parallel metal plate samples were used. The former was used for temperatures above 40 °C, whereas the latter was utilized on test samples below 40 °C. To this end, the frequencies of between 0.1 and 50 Hz and preselected angular deflected (or torque) amplitudes were utilized. The chosen amplitudes were within the linear behavior region with test temperature ranging from −10 to 50 °C.

### 2.10. Multiple Stress Creep Recovery (MSCR) Test

Rutting is recognized and employed as the first failure mechanism in the majority of flexible pavement design scenarios. As one of the key elements used in asphalt mixtures, bitumen or asphalt binder can play a critical role, and altering its properties can considerably postpone or even prevent these failures in some cases. To better determine bitumen performance at high temperatures, the US Federal Highway Agency (FHWA) proposed a multi stress creep recovery (MSCR) test and an unrecoverable accepted parameter. In the present study, the asphalt sample’s elastic response under creep and recovery were determined at 0.1 kPa and 3.2 kPa stress levels at a temperature of 64 °C using the DHR-2 Rheometer. The asphalt sample was tested following ASTM D 7175 specification using a 25 mm parallel plate and a setting of 1 mm gap. Constant stress for 1 s was loaded on the specimen before 9 s recovery. The non-recoverable creep compliance (*J_nr_*) and the difference in *J_nr_* values from the MSCR test results were calculated using Equations (1)–(3).
(1)$$J_{nr}=\frac{\sum_{n=1}^{10}{\left(\frac{\varepsilon_{10}}{\sigma }\right)}_{n}}{10}$$
(2)$$J_{nr\_diff}(\%)=\frac{J_{nr\_3200}-J_{nr\_100}}{J_{nr\_100}}\times 100$$
(3)$$R=\frac{\varepsilon_{p}-\varepsilon_{u}}{\varepsilon_{p}}\times 100\%$$
where *J_nr_* is the non-recoverable creep compliance and *J_nr_diff_* is the difference between the 0.1 kPa and the 3.2 kPa stress level, respectively. *ε_p_* represents the peak strain, and *ε_u_* represents the unrecovered strain.

## 3. Results and Discussion

The following section presents the results and discussion of laboratory experiments conducted that examine the effect of nano silica on the properties of TB rubberized asphalt binders and its reaction/modification mechanism.

### 3.1. Effect of Nano Silica on Homogeneity of TB Rubberized Modified Bitumen by AFM

AFM is a robust and powerful technique for studying the surface morphology of asphalt. One of the advantages of AFM is that samples do not require prior treatment; hence, the morphology can be directly observed without alteration as a result of cooling, heating, stretching or the use of solvents, which significantly alter the colloid structure of asphalt. Figure 4a–d present AFM images of control and nano-SiO_2_ binders, which also demonstrate the appearance and disappearance of bee-like structures.

According to the extant literature, bee-like structures are dispersed throughout surrounded by a soft homogenous matrix that is attributed to the presence of wax and asphaltenes. However, Figure 4d shows that the bee-like structures disappeared with 3% nano-SiO_2_ content in asphalt. This indicates that the light components of asphalt were absorbed by nano silica particles, which prevent aggregation. Furthermore, the nano-SiO_2_ was evenly distributed in asphalt, which inhibits the clustering of asphaltene in bitumen and culminated in an even micro-morphology.

### 3.2. Characterization of TB Rubberized/Nano Silica Composite Modified Asphalt by XRD

The asphalt binders modified with nano silica particles were evaluated by XRD. As shown in Figure 5, we observed that modified asphalt with different contents of nano silica has no sharp peaks in their pattens, which indicates that the materials were amorphous and did not have a crystalline structure. Furthermore, a broad peak near 2-Theta = 18° for all asphalt binders is ascribed to the asphaltenes structure [26]. A strong broad peak of nano silica is at 23°, which corresponds to amorphous nano silica particles. In other words, it confirms the presence of nano silica and proper dispersion in asphalt matrix.

### 3.3. Effect of Nano Silica on Mechanism of TB Rubberized Asphalt by FTIR

FTIR is a powerful approach that is used to identify the polymer modifiers in rubber-nano silica modified bitumen as well as the determination of the modification mechanism of the latter. Herein, Figure 6a,b presents the FTIR spectrum of PDMS modified nano silica and TB rubberized/nano silica-modified composite asphalt binders. For PDMS modified nano silica powder, the strong absorption peaks lying in the ranges of 950–1300 cm^−1^, 750–900 cm^−1^, and 400–530 cm^−1^ are ascribed to the antisymmetric stretching vibration of Si–O–Si, deformation vibration symmetric stretching vibration and the deformation vibration of O–Si–O, respectively [27]. It is worth noting that PDMS surface-treated nano silica did not have OH groups on its surface; hence, it could promote nano silica–polymer interaction instead of nano silica–nano silica aggregation, which comes with the presence of OH groups in untreated nano silica [25].

For TB rubberized/nano silica modified composite asphalt binders, the peaks around 2850 cm^−1^ as well as 2920 cm^−1^ are ascribed to the aliphatic chain’s C-H stretching vibrations [25]. The 1600 cm^−1^ band correlates with the aromatics (stretching C=C aromatic) [28,29,30]. The aliphatic index band (bending C-H of -(CH_2_-) _n_-) and aliphatic branched band (bending C-H of CH_3_) are observed at 1457 cm^−1^ and 1374 cm^−1^. The absorption peaks around 1031 cm^−1^ were S=O stretching vibrations [25], whereas those at 724 cm^−1^ and 810 cm^−1^ were C-H bending vibrations in Aromatics [30]. No absorption peaks attributed to rubber particles were observed in case of rubber modified asphalt. Hence, the interaction between rubber and asphalt was physical. However, the modification mechanism between rubberized asphalt and nano silica was chemical. Yao et al. attributes the reaction between asphalt binder and nano silica to the -OH groups of the nano silica [31], but this was not likely, since PDMS surface-treated nano silica was employed in this study. Liang et al. observed that microwave irradiation on CR particles results in the formation of groups containing oxygen due to the existence of air during microwave treatment [32].

When the chemical bond is cleaved, the oxygen atoms are inevitably linked. After mixed with asphalt, the microwave activation would affect the properties of CR, reducing the aging index. MCR swelled to the point where TB rubberized asphalt established internal network structures with nano silica, limiting oxygen infiltration. The lightweight components’ volatilization was minimized during the aging process, and the density and structural stability of the asphalt molecules were improved. Microwave activation raised CR’s specific surface area and surface activity even more. The lightweight components were absorbed and the anti-aging components were released by the MCR particles, which swelled and disintegrated in the asphalt. As a result, the anti-aging abilities of TB rubber/nano silica composite modified asphalt was enhanced. Hence, microwave irradiation on CR particles lead to the aging of CR particles. 

### 3.4. Evaluation of Storage Stability of TB Rubberized Asphalt Modified with Nano Silica

The major drawback of using crumb rubber modified asphalt is poor storage stability. CR modifiers often separate from modified bitumen during hot storage and transportation, thereby obliterating the very essence of asphalt modification. As an alternative, TB rubberized asphalt is employed for improved storage stability. However, because of indigestible particles in TB modified asphalt, phase separation still persists. In the present study, a method proposed by Wang et al. utilizing segregation and MSCR tests was utilized to evaluate the stability index of TB samples [33]. In this method, samples are first subjected to segregation test for 48 h. Thereafter, samples from the top, middle and bottom samples are cut from the cigar tubes and subjected to MSCR testing. This method is premised on the understanding that MSCR parameters are sensitive to microstructure changes of modified bitumen and that any changes due to phase separation between the top, middle, and bottom of cigar test samples can easily result in microstructure changes in MSCR values. Accordingly, the top, middle and the bottom of cigar samples from segregation tested samples were subjected to MSCR testing. The segregation index (SI) was determined using Equation (4):(4)$$SI=\frac{\left|J_{nr.t}-J_{nr.b}\right|}{J_{nr.average}}$$
where $J_{nr.b}$ and $J_{nr.t}$ are values corresponding to the lower and upper portions of the cigar tube. $J_{nr.average}$ is the average $J_{nr}$ values of the three equally cut portions of the tube.

Higher values of SI indicate poor storage stability, whereas a small value of SI indicates better storage stability of the sample. Table 5 presents the segregation index (SI) of TB rubberized asphalt with and without nano silica modification. The results of this work found that all TB rubberized has good storage stability. However, nano silica enhanced the storage stability of the binders further. The improved storage stability was attributed to the nano silica and rubber particle that arise because of the PDMS or silicone oil chains in silicone oil modified nano silica bonding with rubber chains physically. These bonds result in the immobilization of rubber chains, which leads to slower movements in the vicinity of nano silica particles, hence preventing a phase separation by bonding the asphalt samples and CR particles together [34].

### 3.5. Master Curve of Complex Modulus

The use of master curves to understand how binder type and chemical make-up affect the viscoelastic behavior of binders is a beneficial strategy [35]. It is possible to obtain interpolated values of property for any combination of temperature and frequency within the measurement range after building the master curve. Furthermore, laboratory results may be compared to those from different test conditions such as frequencies and temperatures.

The complex modulus from a DSR frequency sweep test was utilized to generate the master curve following the Time–Temperature Superposition Principle (TTSP). This process could be achieved by utilizing the Christensen–Anderson–Marasteanu (CAM) model described by the author elsewhere [25]. This process could be achieved by utilizing the Christensen–Anderson–Marasteanu (CAM) model given in Equations (5)–(7):(5)$$G^{*}=G_{e}^{*}+\frac{G_{g}^{*}-G_{e}^{*}}{{\left[1+{\left(f_{c}/f^{\prime }\right)}^{k}\right]}^{m_{c}/k}}$$
(6)$$f^{\prime }=f\times \alpha_{T}$$
(7)$$\log\;\alpha_{T}=\frac{C_{1}(T-T_{ref})}{C_{2}+(T-T_{ref})}$$
where $f^{\prime }$ is the reduced frequency of complex modulus $G^{*}$, while when the frequency is infinitely near to zero, it is the equilibrium complex modulus $G_{e}^{*}$, and binders are usually zero. $G_{g}^{*}$ is the glass complex modulus when the frequency approaches infinity. $f_{c}$ is the parameter related to location also known as crossover frequency. $f^{\prime }$ represents the reduced frequency, while f is the original frequency. k and $m_{e}$ are the shape parameters. Fitting the William–Landrel–Ferry (WLF) function yielded the shift factors $\alpha_{T}$. $T_{ref}$ is the reference temperature; $C_{1}$ and $C_{2}$ are the model parameters.

Figure 7 presents a master curve of complex modulus for the unaged sample of control and modified asphalt with nano silica. It can be seen that the addition of nano silica particles to control binders results in changes in rheological properties almost similar to filler materials. The addition of nano silica to control binders increases the complex modulus across the loading frequency and temperatures. The higher complex modulus is attributed to the reinforcing effect of nano silica on asphalt binders and also the high specific surface area, which leads to more interaction between nano silica particles and rubberized asphalt. In higher loading frequency, the complex modulus of control and modified binders reached a glass modulus constant value, and the influence of nano silica becomes less manifest. Originally, a more elastic behavior of asphalt is preferred in its performance against fatigue cracking and rutting but can be detrimental to low-temperature cracking.

### 3.6. Rheological Aging Index (RAI)

During the aging process of asphalt, some light and volatile components of asphalt evaporate from asphalt, leaving it brittle, which results in pavement cracking. Therefore, asphalt binders with lower oxidative aging potential are desirable [36]. In this research, we examine the impact of nano silica on the aging susceptibility of terminal blend rubberized asphalt using the rheological aging index (RAI) based on Equation (8).


(8)
$$RAI=\frac{G_{Aged\;}^{*}}{G_{Unaged\;}^{*}}\exp\left(\delta_{Aged\;}-\delta_{Unaged\;}\right)$$


The input parameters for Equation (8) are presented in Table 6. The rheological test was performed at 25 °C and 10 rad/s on both the control binders and those modified by nano silica in conformity with ASTM D7175. Figure 8 presents the results of the ageing performance tests of control and nano silica modified binders.

It can be seen clearly from Figure 8 that RAI decreased with the increase in nano silica content. Lower values of RAI indicate an improved aging performance of modified bitumen. Fini et al. ascribed nano silica’s anti-aging benefits to its high surface area, large pore size, and the presence of hydroxyl groups on the surface of the nano particles [37]. Fini et al. went on to suggest that the oxygen atom in the hydroxyl groups on nano-surface silica’s is far more electronegative than the hydrogen atom. As a result, it forms hydrogen bonds easily with aromatic sheets in polar aromatics and asphaltene molecules. This situation is unlikely in our study, since nano silica is surface treated with PDMS, which is methyl terminated, and there is no hydroxyl group on the nano silica surface (Aerosil R202). We therefore concluded that the activation of CR particles possesses many anti-oxidant components [38]; therefore, it was generally expected that the anti-aging index RAI of the binder would be generally low.

More importantly, nanomaterials migrate to the surface of composite materials and serve as a barrier to protect host polymers, as is the case for the nano silica (SiO_2_)/rubber system [39]. With the increase in the content of nano silica, oxidative aging reduces considerably. As a result of delayed aging, asphaltenes hydrogenate into polycyclic aromatics or hydroaromatic and the aromatic compounds increase, which makes the asphalt binder soft. Oxidative aging makes asphalt brittle and stiffer due to the increased resins and quasi-solid asphaltenes and the decreased amount of soft aromatic components, leading to increased binder stiffness [40].

### 3.7. Effect of Nano Silica on High-Temperature Properties of TB Rubberized Asphalt by MSCR

The MSCR test may be used to determine the rutting resistance of binders. The test is carried out in order to better understand the behavior of binder materials when subjected to high pressures outside of the linear viscoelastic area. The improvement brought about by modification has been evaluated using non-recoverable creep compliance, which is likewise more sensitive to the stress dependency characteristics of modified binders. Figure 9a,b presents the effect of nano silica on rutting performance of TB asphalt binders. Therein, the non-recoverable creep compliance (*J_nr_*) was used to analyze MSCR test results for both 3.2 kPa and 0.1 kPa stress levels. The results demonstrated that the addition of nano silica into TB rubberized asphalt significantly decreases the *J_nr_* values indicating an improved rutting resistance. When the results were compared at two different stress levels, the *J_nr_* values at 3.2 kPa were somewhat lower than the value at 0.1 kPa. It can also be concluded that when the content of nano-silica increases, the creep recovery at the end of each cycle increases. This means that more delayed elastic deformation returns to its initial state within 9 s of each cycle’s rest time. 

On the other hand, the elastic behavior of asphalt binder under loading is determined by the percentage recoverable strain. It is a crucial criterion for determining an asphalt binder’s capacity to rebound after deformation. The lesser the sensitivity to permanent deformation, the larger the proportion of recoverable strain in the asphalt binder. In other words, the higher the recovery percentage (R%), the better the bitumen elastic characteristics, which occurs because more strain returns to its initial form within 9 s of the rest interval in each cycle. Generally, non-recoverable creep compliance has an inverse relationship with percent recoverable strain. Figure 9c presents results of the percentage recoverable (R%) of TB asphalt samples. As can be seen, adding nano silica to TB rubberized asphalt binder improves percentage recovery (R%) for both original and short-term aged samples. When comparing the TB rubber/nano silica composite modified binders to the TB rubberized control binder, it can be noted that the former has a greater percent recoverable strain value. Additionally, increasing the content of nano silica results in increased R% which indicate improved elastic properties or delayed deformation. Comparing the original samples with aged samples, R% results show that the aged samples had higher strain recovery. Nano silica improves the aging properties of TB binders and therefore has a higher strain recovery. This implies that adding tiny amounts of nano silica to asphalt binder increases its elastic recovery and rutting resistance.

Figure 9d confirms that all *J_nr_diff_* values were less than 75%, which satisfy the standard specifications. Table 7 presents the results of MSCR test and illustrates the corresponding traffic values each binder can carry herein referred to as Standard (S), Heavy (H), Very heavy (V), or Extremely heavy (V) to account for traffic volume (E). Findings revealed that TB rubberized bitumen containing 3% nano silica (weight of asphalt) correlated with very heavy (V) traffic greater than 10 million ESALs at a relatively slow speed of 12.4 mph (20 km/h). Meanwhile, the control group of TB rubberized bitumen without nano silica modification could only carry heavy (H) traffic greater than 3 million ESALs at speeds between 12.4 and 43.4 mph. TB rubberized samples with nano silica exhibit increased modulus compared to control binders, which is beneficial to binder resistance against permanent deformation. This is attributed to the reinforcing effect of nano silica on rubberized asphalt matrix [41]. Recent studies also found an increase in complex modulus for fumed silica/poly(butylene succinate) composite blend after increasing the content of filler [34]. This was due to the creation of physical network structures that reduce the movements of polymer chains and increase mechanical properties. In another study, Liao et al. concluded that there exist strong interfacial interactions between montmorillonite and polybutadiene rubber, resulting in good dispersion, which also functions as “pseudo-crosslinking” points. Past studies concluded that the increase in complex modulus is due to the reinforcing effect of nano silica on rubber particles as well the entanglement of surface PDMS groups of modified nano silica with CR matrix chains [34]. We therefore concluded that the reinforcing effect of nano silica on TB rubber asphalt and the polymeric network structures increased the binder stiffness and decreased non-recoverable permanent strains.

## 4. Conclusions

It has been primarily reported in the published literature that TB rubberized asphalt binders exhibit inferior high-temperature rutting performance. Rutting is identified and used as the first failure mechanism and most important design criteria of flexible pavement. The main cause of rutting in asphalt pavement has been recognized as “accumulated strain”, which is caused by traffic loading. Although, the addition of nanomaterials, particularly nano silica, has been reported to improve the durability and performance of crumb rubber modified asphalt essentially, the performance improvement and reaction mechanism studies have not been extended to TB binders. Moreover, the modification or reaction mechanism has not been clearly understood. Consequently, this research evaluated the high-temperatures properties and reaction mechanism of terminal blend (TB) rubber/nano silica composite modified asphalt using microwave-activated rubber. To this end, laboratory tests such as TFOT and PAV aging tests, segregation test, frequency sweep, MSCR, FTIR, FTIR and AFM were conducted using prepared samples. Based on these rigorous tests, the following conclusions were made:The addition of nano silica significantly improved the rheological properties, storage stability and high-temperature rutting resistance of terminal blend rubberized asphalt. This is attributed to the reinforcing effect of nano silica on TB rubber asphalt and its interaction, which leads to the formation of polymeric network structures with PDMS chains intertwined with rubber particles, thereby reducing the movements around the particles and hence reducing phase separation. In addition, the polymeric network structure increases the binder stiffness and decreases nonrecoverable permanent strains, which is beneficial to reducing the permanent deformation of asphalt binders.The rheological aging index (RAI) on PAV samples revealed that nano silica improves the aging of asphalt. This aging improvement of TB samples with nano silica is due to the interaction of polymer chains and nano silica particles which could delay the aging of polymers and raise the decomposition temperature. In addition, nanosilica migrates to the surface of composite materials and serve as a barrier to protect host polymers, resulting in improved anti-aging properties of the binders.The incorporation of nano silica into TB rubberized binders improves material compatibility between SiO_2_ and TB binders, which is related to two factors: the manufacturer’s surface pretreatment of fumed silica, which inhibited particle–particle (SiO_2_-SiO_2_) interaction while increasing particle–polymer interaction and the microwave activation of CR resulting in a loose and porous surface, which increased the reaction area of CR with asphalt binders.Based on XRD results, the TB-rubberized asphalt with different contents of nano silica had no sharp peaks in their pattens, which indicates that the materials were amorphous and did not have a crystalline structure.AFM images showed bee-like structures which eventually disappeared after the addition of 3% Nano-SiO_2_, indicating that the light components of asphalt were absorbed by nano silica particles, which prevent aggregation.The reaction mechanism between asphalt and rubber particles was physical, owing to the absence of new absorption peaks in the IR spectrum, whereas the interaction between TB binder and nano silica was chemical.

Despite the performance improvement of TB rubber/nano silica composite modified asphalt, this research did not examine the effect of activated and unactivated CR particles on TB rubberized/nano silica composite modified asphalt. Further research can examine the influence of microwave activation of rubber on the performance of TB rubber/nano silica composite modified asphalt.

## Figures and Tables

**Figure 1 nanomaterials-12-04388-f001:**
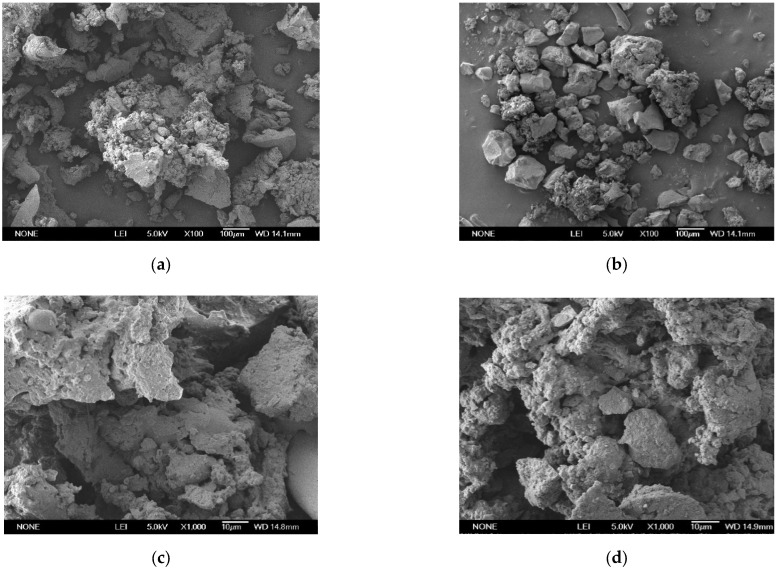
Scanning electron microscopy (SEM) images. (**a**) unactivated rubber (×100 magnification); (**b**) microwave activated (×100 magnification); (**c**) unactivated rubber (×1000 magnification); (**d**) microwave activated (×1000 magnification).

**Figure 2 nanomaterials-12-04388-f002:**
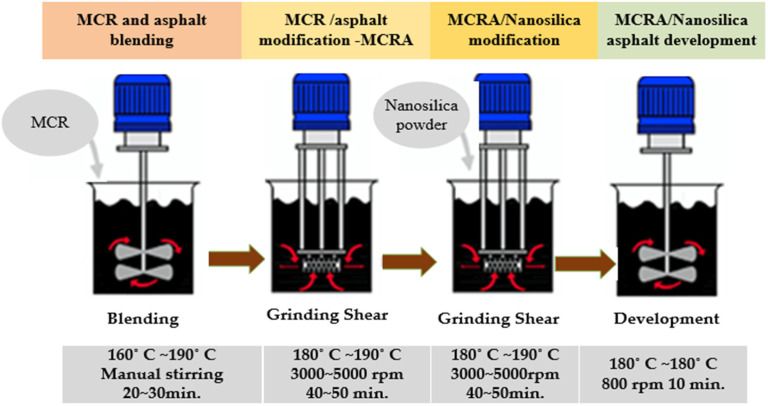
Modification procedure for laboratory preparation of TB rubber/nano silica composite modified asphalt.

**Figure 3 nanomaterials-12-04388-f003:**
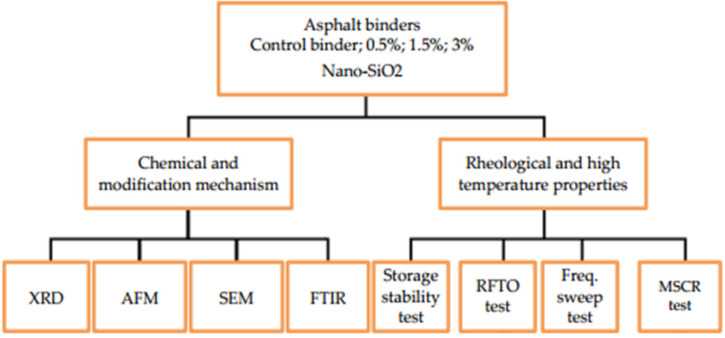
Experimental flow chart utilized herein.

**Figure 4 nanomaterials-12-04388-f004:**
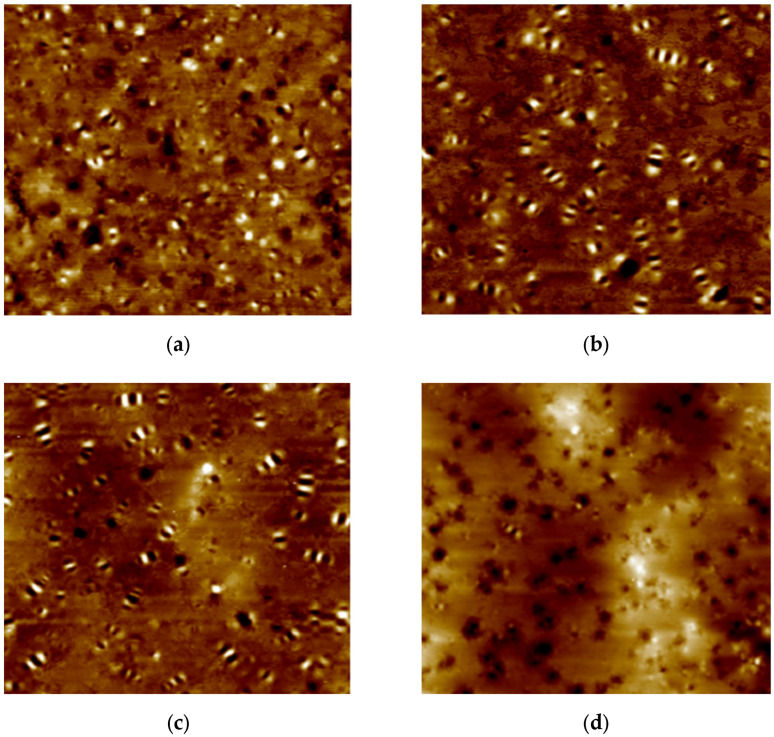
AFM images of TB rubberized asphalt (**a**) Control binder; (**b**) 0.5% Nano-SiO_2_; (**c**) 1.5% Nano-SiO_2_; (**d**) 3% Nano-SiO_2_.

**Figure 5 nanomaterials-12-04388-f005:**
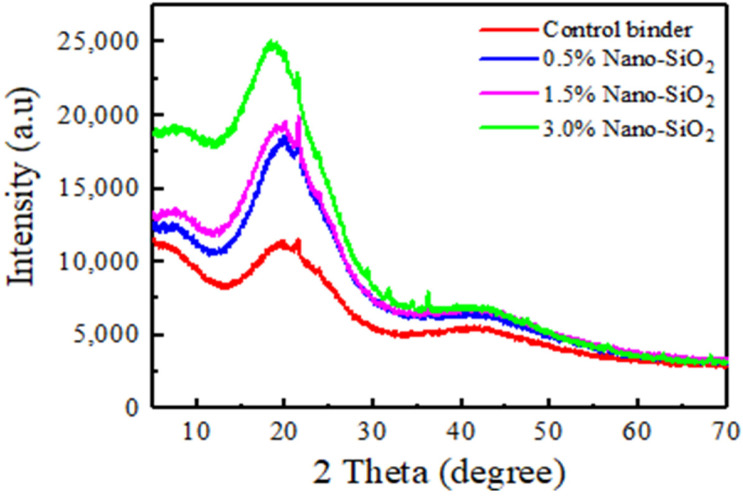
XRD patterns of terminal blend rubberized asphalt with and without nano silica particles.

**Figure 6 nanomaterials-12-04388-f006:**
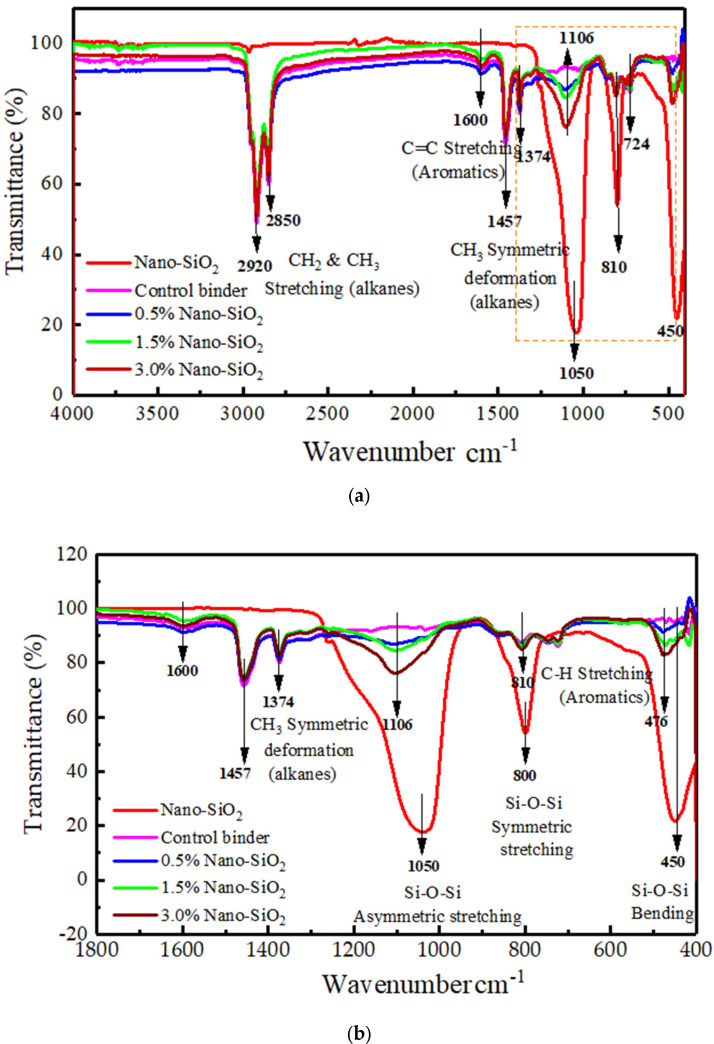
FTIR spectra of crumb rubber/nano silica composite pavement: (**a**) all functional groups, (**b**) detailed finger printing region.

**Figure 7 nanomaterials-12-04388-f007:**
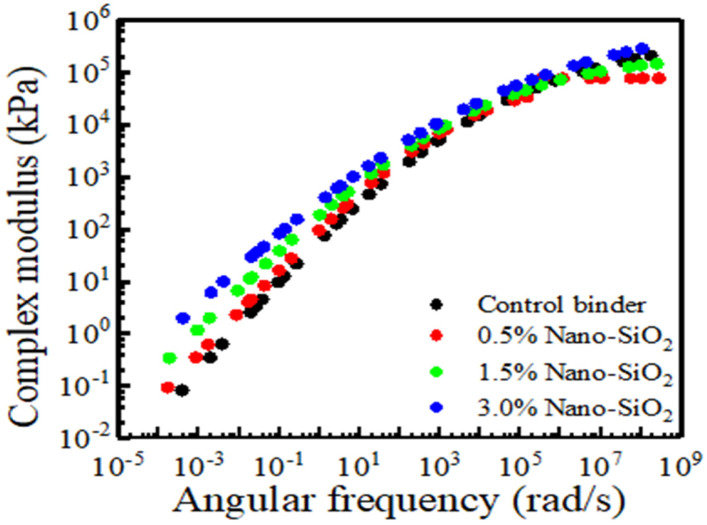
Master curve of complex modulus of TB rubberized asphalt binder.

**Figure 8 nanomaterials-12-04388-f008:**
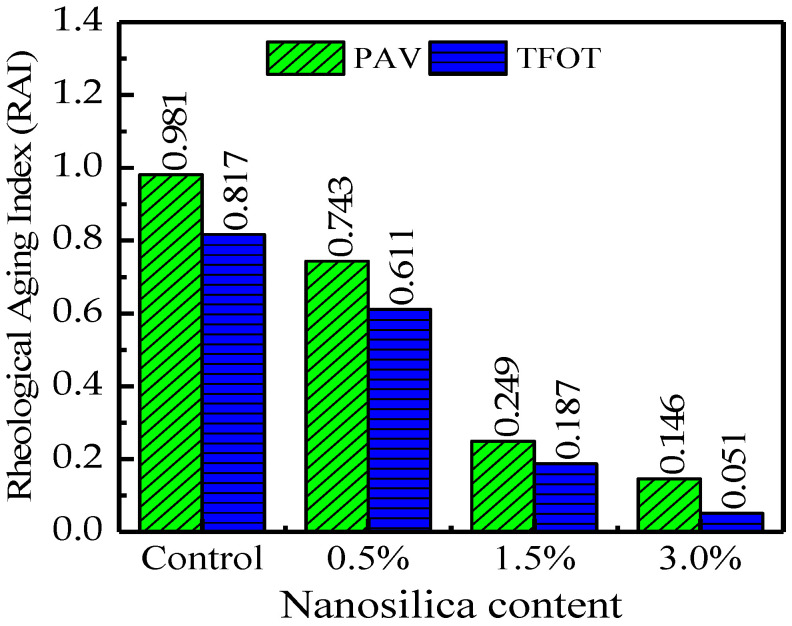
Rheological aging index terminal blend rubberized asphalt with different contents of nano-silica.

**Figure 9 nanomaterials-12-04388-f009:**
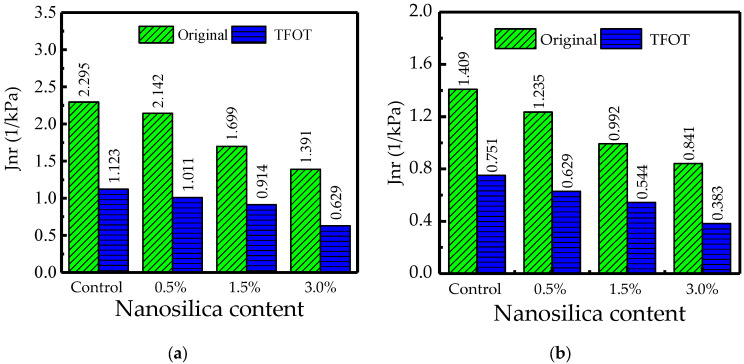
Multiple stress creep recovery (MSCR) of asphalt samples. (**a**) *J_nr0.1_*; (**b**) *J_nr3.2_*; (**c**) percentage recoverable; (**d**) *J_nr_* Difference.

**Table 1 nanomaterials-12-04388-t001:** Fundamental properties of bitumen penetration grade 80–100.

Test Items	Values	Criteria	Standard
Softening point (°C)	46	≥45	T 0606-2011
Penetration at 25 °C (0.1 mm)	84	80~100	T 0604-2011
Density at 15 °C (g/cm^3^)	1.03	-	T 0603-2011
Ductility at 15 °C (cm)	>100	≥100	T 0605-2011
Ductility at 10 °C (cm)	>100	≥20	T 0605-2011
Dynamic viscosity at 60 °C (Pa∙s)	178	≥160	T 0620-2011
Content of wax (%)	1.9	≤2.2	T 0615-2011
Solubility (%)	99.84	≥99.5	T 0607-2011
Flashpoint (°C)	>300	≥245	T 0611-2011
Residue after RTFOT:			
Mass change (%)	0.112	−0.8~0.8	T 0609-2011
Residual penetration (%)	62.4	≥57	T 0609-2011
Residual ductility (%)	11.9	≥8	T 0609-2011

**Table 2 nanomaterials-12-04388-t002:** Fundamental properties of un-activated crumb rubber particles.

Properties	Values	Criteria
Ash content (%)	4.2	≤8
Density g/cm^3^	1.18	1.10~1.30
Content of metals (%)	0.03	<0.05
Moisture content (%)	0.4	≤1
Content of carbon black (%)	30	≥28
Content of rubber hydrocarbon (%)	48.0	≥42
Acetone extracts (%)	8	≤22

**Table 3 nanomaterials-12-04388-t003:** The physical properties of fumed silica, herein PDMS modified nano silica.

Material Type	Specific Surface Area (m^2^/g)	Density g/cm^3^	Particle Size (nm)	Carbon (%)	SiO_2_ (%)
Aerosil R202 fumed silica	80~120	0.04	14	3.5~5	≥99.8

**Table 4 nanomaterials-12-04388-t004:** List of asphalt binder utilized herein.

Labels	Modifier	Definition and Description of Asphalt Binders
MCRA	Microwave-activated crumb rubber (MCR) modifier	MCR without nano silica control group
0.5% SiO_2_	MCR and Nano silica	Blended MCR with 0.5% nano-SiO_2_ (wt % asphalt)
1.5% SiO_2_	MCR and Nano silica	Blended MCR with 1.5% nano-SiO_2_ (wt % asphalt)
3.0% SiO_2_	MCR and Nano silica	Blended MCR with 3.0% nano-SiO_2_ (wt % asphalt)

**Table 5 nanomaterials-12-04388-t005:** Segregation index of TB rubberized asphalt with and without nano silica.

Binder	$J_{nr.top}$	$J_{nr.botton}$	$J_{nr.\;average}$	Segregation Index (SI)
Control binder	0.602161	0.949826	0.775994	0.448025
0.5% nano-SiO_2_	1.872696	1.586555	1.729625	0.165435
1.5% nano-SiO_2_	2.103066	2.287947	2.361135	0.082143
3.0% nano-SiO_2_	3.307155	3.484055	3.501161	0.052097

**Table 6 nanomaterials-12-04388-t006:** Complex modulus for original and aged samples.

Asphalt Samples	Complex Modulus (kPa)		
	Original	RTFO	PAV
Control binder	1.37649	1.57664	2.30366
0.5% Nano-SiO_2_	1.46518	2.01542	2.60868
1.5% Nano-SiO_2_	1.77617	2.14168	2.86124
3.0% Nano-SiO_2_	1.98015	2.20525	3.36264

**Table 7 nanomaterials-12-04388-t007:** Traffic grading according to *J_nr_* values.

Max. *J_nr_* 3.2 (1/kPa)	Temperature (°C)	Max. *J_nr_*__diff_ (%)	Grade	Traffic Level	
<4.0	64	75	Standard (S)	<3 million ESALs	>43.4 mph
<2.0	64	75	Heavy (H)	>3 million ESALs	12.4~43.4 mph
<1.0	64	75	Very heavy (V) grade	>10 million ESALs	<12.4 mph
<0.5	64	75	Extreme (E)	>30 million ESALs	<12.4 mph

## Data Availability

The authors pledge to make data available when needed.

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
