# Peer review of "Evaluating the High-Temperature Properties and Reaction Mechanism of Terminal Blend Rubber/Nano Silica Composite Modified Asphalt Using Activated Rubber"

_nanomaterials, 2022, doi:10.3390/nano12244388_

Round 1

Reviewer 1 Report

Comments:

The article titled "Evaluating the High Temperature Properties and Reaction 2 Mechanism of Terminal Blend Rubber / Nano Silica Composite 3 Modified Asphalt Using Activated Rubber" is in my opinion well written and well done.

It clearly describes how the addition of nanoscale silica (aerosil) modifies the physico-chemical properties of an asphalt.

I therefore find it generally a job well done

Personally I know aerosil from having used it in some laboratory tests, and I am sure that what is described corriposda to the truth.

I have only a few points to raise:

- erosil is a nanometric amorphous silica

In figure 7 (FT-IR) it is well explained, and it is evident that as the silica increases, there is a greater instrumental signal.In fact, the silica band at 1050 cm-1 varies as the aerosil concentration varies (greater intensity band = higher percentage of silica).

Figure 6 shows the diffractogram (XRD) of the same formulations.

Aerosil is an amorphous substance, therefore, no maximum diffraction should be seen.

Furthermore, as the concentration of silica increases, the amorphous nature of the compound obtained should be more evident.

Can the authors explain why as silica increases the amorphous nature of the final compound decreases (the classic amorphous curve highlighted by the XRD analysis decreases as silica increases instead of growing)?

Also, what do the authors attribute the 21 °-degree peak visible in the difractogram?

Finally, it is evident that a greater concentration of nanometric silica makes the resulting asphalt more compact.

Do the authors believe that this could have practical applications of use?

Furthermore,

if the journal is called nanomaterials, I believe it is important to show the nano-metric nature of the result of the experiments described. I think it is therefore useful, for the purposes of greater understanding, to insert some TEM images to observe the nanometric nature of what has been done. Obviously, I also consider it important to add the description of the TEM microscope used in the "materials and methods" section

Reviewer 2 Report

The proposed paper looks more like a technical report than a real scientific article. I think that there is no need to put a picture of the equipment unless you developed a homemade apparatus (which does not seem to be the case here). Why re-explain some classical experimental techniques in the paper?

The fact of adding nanoparticles is well known to modify/improve the mechanical properties  ... what is the added value of this work to the field?

It is extremely difficult for the potential reader to have a clear view of what you did and how he/she can extend to other CRs since they are all different for practical reasons (coming from used tires).

Try to reformulate the paper and extend the discussions and explain how the reader can extract useful information !

At this stage, it does not fit with the standards for a publication in Nanomaterials!

Round 2

Reviewer 2 Report

Nice improvement of the paper